# The Role of TPM3 in Protecting Cardiomyocyte from Hypoxia-Induced Injury via Cytoskeleton Stabilization

**DOI:** 10.3390/ijms25126797

**Published:** 2024-06-20

**Authors:** Ke Huang, Weijia Yang, Mingxuan Shi, Shiqi Wang, Yi Li, Zhaoqing Xu

**Affiliations:** 1Key Laboratory of Preclinical Study for New Drugs of Gansu Province, School of Basic Medical Sciences, Lanzhou University, Lanzhou 730030, China; huangk18@lzu.edu.cn; 2Key Laboratory of Dental Maxillofacial Reconstruction and Biological Intelligence Manufacturing, School of Stomatology, Lanzhou University, Lanzhou 730030, China; yweijia2023@lzu.edu.cn (W.Y.); shimx2021@lzu.edu.cn (M.S.); wangshq2020@lzu.edu.cn (S.W.)

**Keywords:** TPM3, hypoxia, cardiomyocyte injury, oxidative stress, cytoskeleton

## Abstract

Ischemic heart disease (IHD) remains a major global health concern, with ischemia-reperfusion injury exacerbating myocardial damage despite therapeutic interventions. In this study, we investigated the role of tropomyosin 3 (TPM3) in protecting cardiomyocytes against hypoxia-induced injury and oxidative stress. Using the AC16 and H9c2 cell lines, we established a chemical hypoxia model by treating cells with cobalt chloride (CoCl_2_) to simulate low-oxygen conditions. We found that CoCl_2_ treatment significantly upregulated the expression of hypoxia-inducible factor 1 alpha (HIF-1α) in cardiomyocytes, indicating the successful induction of hypoxia. Subsequent morphological and biochemical analyses revealed that hypoxia altered cardiomyocyte morphology disrupted the cytoskeleton, and caused cellular damage, accompanied by increased lactate dehydrogenase (LDH) release and malondialdehyde (MDA) levels, and decreased superoxide dismutase (SOD) activity, indicative of oxidative stress. Lentivirus-mediated TPM3 overexpression attenuated hypoxia-induced morphological changes, cellular damage, and oxidative stress imbalance, while TPM3 knockdown exacerbated these effects. Furthermore, treatment with the HDAC1 inhibitor MGCD0103 partially reversed the exacerbation of hypoxia-induced injury caused by TPM3 knockdown. Protein–protein interaction (PPI) network and functional enrichment analysis suggested that TPM3 may modulate cardiac muscle development, contraction, and adrenergic signaling pathways. In conclusion, our findings highlight the therapeutic potential of TPM3 modulation in mitigating hypoxia-associated cardiac injury, suggesting a promising avenue for the treatment of ischemic heart disease and other hypoxia-related cardiac pathologies.

## 1. Introduction

Ischemic heart disease (IHD) is a prevalent cardiovascular condition characterized by myocardial ischemia and hypoxia due to coronary artery insufficiency, leading to significant morbidity and mortality worldwide [1,2]. Despite advancements in therapeutic interventions such as reperfusion strategies, ischemia-reperfusion injury (IRI) remains a major concern in IHD management, exacerbating myocardial damage and potentially leading to adverse cardiac events, including myocardial infarction and heart failure [3]. During IRI, a cascade of intracellular molecular events occurs, including dysregulation of ion homeostasis and generation of reactive oxygen species (ROS) [4,5,6,7,8,9,10,11]. Additionally, mitochondrial permeability transition pore (mPTP) opening contributes to mitochondrial dysfunction and cell death [12,13,14,15]. Among the manifestations of myocardial injury, cell swelling or edema has emerged as a critical pathological process, indicative of early myocardial ischemia [16,17]. Alterations in cell volume are regulated by osmotic pressure, ion homeostasis, and water channel proteins [18,19,20,21,22,23,24]. Furthermore, the cytoskeleton, composed of microfilaments, microtubules, and intermediate filaments, plays a crucial role in maintaining cell morphology, regulating signal transduction, and modulating cellular responses to external stimuli [25,26,27]. Dysregulation of the cytoskeleton has been implicated in affecting cell structure, signaling pathways, and edema formation, potentially contributing to myocardial ischemic injury [28,29,30,31,32]. Understanding the intricate mechanisms underlying myocardial ischemic injury, including the interplay between cytoskeletal remodeling, cell swelling, and molecular events, is essential for developing effective therapeutic strategies for IHD.

The cytoskeleton, a crucial structural framework within cells, primarily comprises microfilaments, microtubules, and intermediate filaments. This network maintains cell morphology and facilitates cellular processes through intricate interaction networks [25]. Microfilaments, mainly composed of actin, form stable structures that support various cellular functions. These include providing mechanical support, maintaining cell shape, sensing the external environment, regulating cell division, and influencing cell motility [33,34,35]. The functionality of the microfilament cytoskeleton is regulated by actin and other cytoskeleton-associated proteins and signaling pathways [36,37]. Tropomyosin (TPM) serves as a key regulatory factor in the dynamic regulation of the microfilament cytoskeleton. Different TPM isoforms precisely modulate the dynamic balance of actin filaments by competitively binding with key regulatory factors, affecting their polymerization, depolymerization, and other behaviors [38]. For example, TPM3, a member of the TPM family, has been shown to play a crucial role in regulating the stability of the cellular microfilament cytoskeleton. Specifically, mutations in TPM3 could affect or disrupt the regulation of myofibril length in muscle cells, which leads to decreased muscle fiber contraction force and contributes to skeletal muscle diseases [39]. Moreover, TPM3 can regulate the stability of actin filaments and increase cellular stiffness [40,41]. The pathological processes of cardiovascular diseases involve myocardial ischemia, hypoxia, and myocardial cell edema [42,43]. Proper regulation of myocardial cell volume is essential for preventing the occurrence and development of cardiovascular diseases [44,45]. Changes in cellular mechanical properties are important indicators of pathological alterations [46,47,48]. Furthermore, myocardial cell edema involves changes in gene expression profiles, accompanied by alterations in cellular morphology and mechanical properties [49]. Therefore, investigating the mechanisms of TPM3 in myocardial cells and understanding how these regulatory mechanisms affect myocardial cell hypoxia injury is crucial for preventing cardiovascular diseases.

Histone deacetylases (HDACs) have emerged as critical epigenetic regulators in the pathophysiology of cardiovascular diseases. A growing body of evidence implicates HDACs in the modulation of various cellular processes that contribute to the development and progression of CVDs. In atherosclerosis, HDACs are intricately linked with the disease’s progression, and pan-HDAC inhibitors such as scriptaid or trichostatin A (TSA) have demonstrated utility in preventing neointima formation from balloon injury [50,51,52]. Furthermore, HDAC inhibitors have shown promise in attenuating cardiac arrhythmias, potentially through interactions involving the HopX–HDAC2 axis [53,54]. The cardioprotective effects of HDAC inhibitors extend to myocardial infarction, where they have been shown to reduce ischemia-reperfusion injury, improve cardiac function, and minimize infarct size [55,56,57]. Additionally, HDAC1 inhibition protects against hypoxia-induced swelling in H9c2 cardiomyocytes by enhancing cell stiffness [49]. Collectively, these findings underscore the complex yet pivotal role of HDACs in CVDs and highlight the therapeutic potential of HDAC inhibitors in this domain.

Despite the fact that TPM3 can regulate the stability of actin filaments and increase cellular stiffness, and HDAC1 affects cellular rigidity, thereby influencing cardiomyocyte edema, whether TPM3 could be a key molecule in the hypoxic injury process of cardiomyocytes remains unknown. To address this question, in this study, we conducted a comprehensive investigation into the effects of hypoxia on cardiomyocytes and explored the potential protective role of TPM3 and HDAC1 inhibitor MGCD0103 against hypoxia injury. Our aim was to elucidate the complex mechanisms underlying cardiomyocyte responses to hypoxia injury and emphasize the therapeutic potential of TPM3 modulation in mitigating hypoxia-related cardiac damage.

## 2. Results

### 2.1. Cobalt Chloride (CoCl_2_) Enhanced the Expression of HIF-1α in Cardiomyocytes

The groups cultured under normoxia conditions (high-glucose DMEM medium containing 10% FBS) were labeled as the normoxia group, while the groups cultured under hypoxia conditions (serum-free, sugar-free DMEM medium containing a concentration of 400 μM CoCl_2_) for 5 h were labeled as the hypoxia group. After intervention, total protein was extracted from the cells for Western blot (WB) analysis.

As shown in Figure 1, compared to the normoxia group, CoCl_2_ intervention significantly upregulated the expression levels of hypoxia-inducible factor 1 alpha (HIF-1α) in both AC16 and H9c2 cells, indicating a good modeling effect of chemical hypoxia.

### 2.2. Hypoxia Altered the Morphology of Cardiomyocytes and Caused Cellular Damage

AC16 cells were seeded at the same cell density in cell culture plates and incubated until they fully adhered to the substrate and reached a healthy growth state. Cells were then washed twice with a PBS buffer. The cells were divided into normoxia and hypoxia groups, with the former receiving normoxia culture medium and the latter receiving hypoxia culture medium, and then incubated for 5 h to simulate normal oxygen supply and hypoxia conditions, respectively, to observe the growth and morphological changes of the cells. After the intervention, the cells were washed twice with PBS and then observed under a phase-contrast microscope, with images captured.

As shown in Figure 2A, cells in the normoxia group exhibited the typical elongated or polygonal appearance of cardiomyocytes and were closely arranged. In comparison, cells in the hypoxia group showed significant morphological changes and cellular damage, including irregular morphology and detachment from the substrate, when compared to the normoxia group.

### 2.3. Hypoxia Affected the Arrangement of the Cardiac Cell Cytoskeleton

AC16 cells were seeded in confocal-specific cell culture dishes at an appropriate cell density until they fully adhered and exhibited good growth status. The cells were divided into normoxia and hypoxia groups, and the culture medium was replaced accordingly. After intervention for 5 h, immunofluorescence staining for actin filament was performed using phalloidin, and the cell nuclei were counterstained with DAPI. After staining, confocal microscopy was used to capture images with fixed exposure parameters.

The confocal imaging results in Figure 2B,C demonstrate that under normoxia conditions, the cells exhibited good growth status, and the actin filament cytoskeleton was clearly visible and well-organized. Under hypoxia conditions, the fluorescence intensity of the actin filament cytoskeleton was significantly reduced, showing a blurry and fragmented skeletal structure.

### 2.4. TPM3 Protein Showed Significant Colocalization with F-Actin in AC16 Cardiomyocytes

To further validate the subcellular localization of TPM3 protein in cardiomyocytes, immunofluorescence multicolor staining was conducted. DAPI (blue) was used to label the cell nucleus, phalloidin (red) was used to label F-actin, and TPM3 was labeled with primary antibody and Cy5-conjugated secondary antibody (pink). As shown in Figure 3, in AC16 cells, TPM3 protein was mainly localized in the cytoplasm and cell cytoskeleton, exhibiting significant colocalization with F-actin.

### 2.5. Changes in TPM3 Expression Affected the Growth Rate of Cardiomyocytes

Using lentivirus-mediated methods, four stably transfected cell lines of AC16 cells (AC16: OE_NC, OE_TPM3, RNAi_NC, RNAi_TPM3) and H9c2 cells (H9c2: OE_NC, OE_TPM3, RNAi_NC, RNAi_TPM3) were constructed and cultured. Among them, the stably transfected cell line overexpressing TPM3 gene was designated as OE_TPM3, with the corresponding control group using empty vector lentivirus designated as OE_NC; additionally, the stably transfected cell line with TPM3 gene knockdown was designated as RNAi_TPM3, with the corresponding control group using empty vector lentivirus designated as RNAi_NC. When the cells reached optimal growth conditions, samples were collected, and total protein was extracted for Western blot analysis to verify the gene intervention effects of each stably transfected cell line. The experimental results showed that in AC16 cells and H9c2 cells, both TPM3 overexpression and TPM3 knockdown were effective, resulting in significant changes in TPM3 protein expression levels, indicating successful construction of each stably transfected cell line (Figure 4A,B).

The growth curves of various stably transfected cell lines were assessed using the CCK-8 method to examine the effect of TPM3 expression changes. As shown in Figure 4C, in AC16 cells, lentiviral transfection had no significant effect on cell growth rate. Notably, overexpression of TPM3 significantly increased the growth rate, while TPM3 knockdown significantly decreased the growth rate. A similar trend was observed in H9c2 cells (Figure 4D).

### 2.6. TPM3 Overexpression Mitigated Hypoxia-Induced Morphological Changes and Cellular Damage in Cardiac Myocytes

The AC16 cells overexpressing TPM3 (OE_TPM3) and the corresponding control cells (OE_NC) were seeded in cell culture plates at the same cell density. Cells were allowed to adhere and grow until reaching optimal confluence. Afterward, cells were washed twice with PBS. The cells were then divided into normoxia and hypoxia groups and cultured in either normal oxygen or hypoxia conditions for 5 h. After the intervention, cells were washed twice with PBS and observed under a phase-contrast microscope.

As shown in Figure 4E, under normoxia conditions, while both the control group cells and TPM3-overexpressing group cells exhibited a tight arrangement and typical elongated or polygonal morphology, there appeared to be a noticeable trend suggesting potential differences in cell length between the two groups. Specifically, cells in the TPM3-overexpressing group appeared slightly longer compared to those in the control group. However, these differences could not be quantified. Under hypoxia conditions, the control group cells displayed evident morphological changes and cellular damage, characterized by significantly increased intercellular gaps, irregular cell morphology, and extensive cell detachment. In contrast, the TPM3-overexpressing group cells showed milder cellular damage, maintaining their elongated or polygonal morphology, with slight increases in intercellular gaps, as well as only a small number of cells detached from the substrate.

### 2.7. TPM3 Overexpression Enhanced Cardiomyocyte Cytoskeleton under Normoxia and Hypoxia Conditions

To elucidate the effect of TPM3 overexpression on cardiomyocyte cytoskeleton under normoxia and hypoxia conditions, immunofluorescence multicolor staining was performed. DAPI (blue) was used to label the cell nuclei, while phalloidin (red) was used to label F-actin.

As in Figure 5, under normoxia conditions, both the control and TPM3-overexpressing cardiomyocytes exhibited good growth status, with clear and orderly microfilament cytoskeleton structures. Notably, the TPM3-overexpressing group showed higher fluorescence intensity of the microfilament cytoskeleton compared to the control group, with a denser arrangement of the cytoskeleton. Under hypoxia conditions, the fluorescence intensity of the microfilament cytoskeleton in control cardiomyocytes significantly decreased, the cytoskeleton structure became blurred, and fragmentation of the cytoskeleton was observed. In contrast, TPM3-overexpressing cardiomyocytes under hypoxia conditions showed a decrease in microfilament cytoskeleton fluorescence intensity, but the cytoskeleton remained orderly arranged without obvious microfilament cytoskeleton fragmentation.

### 2.8. TPM3 Overexpression Alleviated Hypoxia-Induced Cardiomyocyte Injury and Protected against Hypoxia-Induced Oxidative Stress

When the cell membrane is injured, lactate dehydrogenase (LDH) is released into the culture medium. Therefore, the release level of LDH in the culture medium can be used to assess the extent of cell injury. The results showed that under normoxia conditions in AC16 cells (Figure 6A), TPM3 overexpression (OE_TPM3) slightly reduced the LDH release level compared to the control group (OE_NC). Cell damage significantly increased under hypoxia compared to normoxia. Additionally, TPM3 overexpression significantly attenuated hypoxia-induced cell damage.

In H9c2 cells (Figure 6B), there was no significant difference in LDH release level between TPM3 overexpression (OE_TPM3) and the control group (OE_NC) under normoxia conditions. Cell damage significantly increased under hypoxia compared to normoxia, and TPM3 overexpression significantly alleviated hypoxia-induced cell damage.

Malondialdehyde (MDA), as one of the main products of lipid oxidation in the body, increases when the imbalance of oxidative stress occurred in cells, making it a useful marker for assessing the severity of intracellular oxidative stress. The results from AC16 cells (Figure 6C) showed that under normoxia conditions, TPM3 overexpression (OE_TPM3) had no significant effect on MDA production compared to the control group (OE_NC). However, under hypoxia conditions, MDA levels significantly increased in all groups compared to normoxia, indicating exacerbated oxidative stress due to hypoxia. Furthermore, under hypoxia conditions, TPM3 overexpression significantly reduced the increase in MDA levels induced by hypoxia. In H9c2 cells (Figure 6D), the trend of MDA content changes under normoxia and hypoxia conditions was similar to that in AC16 cells.

Superoxide dismutase (SOD), as a key antioxidant enzyme, plays a crucial role in preventing abnormal oxidative stress in organisms and is therefore often used as an important indicator to evaluate the degree of oxidative-reductive imbalance in cells. In AC16 cells (Figure 6E), under normoxia conditions, there was no significant difference in SOD activity between the control group (OE_NC) and the TPM3 overexpression group (OE_TPM3). Compared to normoxia, the SOD activity of the control group cells significantly decreased under hypoxia conditions, while TPM3 overexpression significantly alleviated the decrease in SOD activity induced by hypoxia. In H9c2 cells (Figure 6F), the trend of SOD activity changes under normoxia and hypoxia conditions was consistent with that in AC16 cells.

### 2.9. TPM3 Knockdown Exacerbated Hypoxia-Induced Cardiomyocyte Injury and Oxidative Stress, While MGCD0103 Attenuated the Process

As shown in Figure 7A, under normoxia conditions, in AC16 cells, TPM3 knockdown (RNAi_TPM3) caused slight cell injury compared to the control group (RNAi_NC), while MGCD0103 treatment (RNAi_TPM3 + MGCD0103) significantly alleviated the cell injury caused by TPM3 knockdown. Under hypoxia conditions, cell injury significantly increased in all groups compared to the normoxia control group. Furthermore, comparing the hypoxia groups revealed that TPM3 knockdown significantly exacerbated hypoxia-induced cell injury, while MGCD0103 reversed this exacerbation effect significantly. In H9c2 cells (Figure 7B), under normoxia conditions, TPM3 knockdown (RNAi_TPM3) to some extent caused cell injury compared to the control group (RNAi_NC). MGCD0103 treatment (RNAi_TPM3 + MGCD0103) partially alleviated the cell injury caused by TPM3 knockdown, but this difference did not reach statistical significance. Under hypoxia conditions, all groups showed significantly increased cell injury compared to the normoxia control group. Additionally, similar to AC16 cells, comparing the hypoxia groups revealed that TPM3 knockdown significantly exacerbated hypoxia-induced cell injury, while MGCD0103 reversed this exacerbation effect.

As shown in Figure 7C, in AC16 cells, under normoxia conditions, there was no significant difference in MDA content among the groups. Compared to the normoxia control group, all groups showed a significant increase in MDA content under hypoxia conditions, indicating exacerbation of oxidative stress due to hypoxia. Comparing the groups under hypoxia conditions revealed that TPM3 knockdown (RNAi_TPM3) significantly exacerbated the increase in MDA content induced by hypoxia, while MGCD0103 treatment (RNAi_TPM3 + MGCD0103) partially alleviated this trend (although this difference was not statistically significant). In H9c2 cells (Figure 7D), similar to the results in AC16 cells, there was no significant difference in MDA content among the three groups under normoxia conditions. Compared to the normoxia control group, all groups showed a significant increase in MDA content under hypoxia conditions. Comparing the groups under hypoxia conditions revealed that TPM3 knockdown significantly exacerbated the increase in MDA content, while MGCD0103 treatment partially attenuated the increase in MDA content induced by TPM3 knockdown (although this difference was not statistically significant).

In AC16 cells (Figure 7E), under normoxia conditions, compared to the control group (RNAi_NC), TPM3 knockdown (RNAi_TPM3) showed a slight decrease in SOD activity, while MGCD0103 treatment (RNAi_TPM3 + MGCD0103) showed a slight increase in SOD activity compared to the TPM3 knockdown group (RNAi_TPM3). Although there were minor fluctuations in SOD activity among the different treatment groups under normoxia conditions, these differences were not statistically significant. Compared to the normoxia control group, SOD activity significantly decreased in the hypoxia control group. Furthermore, comparing the groups under hypoxia conditions revealed that TPM3 knockdown exacerbated the decrease in SOD activity induced by hypoxia to some extent (although this difference was not statistically significant), while MGCD0103 treatment significantly alleviated the decrease in SOD activity induced by hypoxia. In H9c2 cells (Figure 7F), under normoxia conditions, SOD activity significantly decreased in the TPM3 knockdown group (RNAi_TPM3) compared to the control group (RNAi_NC), while TPM3 knockdown combined with MGCD0103 treatment (RNAi_TPM3 + MGCD0103) showed a slight increase in SOD activity compared to the TPM3 knockdown group (RNAi_TPM3) (although this difference was not statistically significant). Compared to the normoxia control group, SOD activity significantly decreased in the hypoxia control group. Additionally, comparing the groups under hypoxia conditions revealed that TPM3 knockdown significantly exacerbated the decrease in SOD activity induced by hypoxia, while MGCD0103 treatment significantly alleviated this process.

### 2.10. The Potential Mechanisms by Which TPM3 Protects Cardiomyocytes from Hypoxia Injury

To preliminarily elucidate the potential mechanisms by which TPM3 protects cardiomyocytes from hypoxia injury, we conducted protein–protein interaction (PPI) network analysis of TPM3 using GeneMANIA and STRING databases. The results from the GeneMANIA database (Figure 8A) revealed that TPM3 interacts with multiple molecules. These molecules include TPM1, TPM2, TPM4, TNNI1, TNNC2, TNNT1, TNNT2, TNNT3, TNF, MYL1, MYL6B, MYBPC1, NEB, TFPT, TCAP, PTPRG, PTPRB, MYH6, and MYL3. The results from the STRING database (Figure 8B) showed that the molecules associated with TPM3 include TPM1, TNNT1, TNNT2, TFG, TNNI3, NTRK1, TNNC1, ACTA1, ACTC1, and TSPAN16.

To further elucidate the functional roles or pathways involving TPM3 and its related proteins, we performed Gene Ontology (GO) and Kyoto Encyclopedia of Genes and Genomes (KEGG) enrichment analysis based on the function annotations of the molecules involved in the PPI network (Figure 8C). The GO/KEGG enrichment analysis revealed significant associations with various biological processes and pathways related to cardiac muscle development and function. Firstly, enrichment was observed in biological processes (BP), such as “cardiac muscle tissue development”, “muscle tissue development”, and “heart contraction”, indicating the involvement of the identified genes in the development and function of cardiac muscle tissue. Additionally, terms like “actin-mediated cell contraction” and “muscle contraction” suggest their roles in regulating muscle contraction processes. Furthermore, the analysis highlighted KEGG pathways relevant to cardiac function, including “Adrenergic signaling in cardiomyocytes”, “Cardiac muscle contraction”, “Dilated cardiomyopathy”, and “Hypertrophic cardiomyopathy”. These pathways are crucial for the regulation of heart function and the pathogenesis of cardiac diseases. At the molecular function (MF) level, enrichment was observed in terms such as “structural constituent of muscle”, “tropomyosin binding”, “actin filament binding”, and “actin binding”. These terms indicate the involvement of the identified genes in the organization and function of the actin cytoskeleton and sarcomeric structures, essential for muscle contraction. Overall, the enrichment analysis suggests that the identified genes play crucial roles in cardiac muscle development, function, and disease pathogenesis, acting through various biological processes and molecular mechanisms.

## 3. Discussion

IHD remains a significant cause of morbidity and mortality worldwide, with IRI exacerbating myocardial damage despite therapeutic interventions. In this study, we comprehensively investigated the effects of hypoxia on cardiomyocytes and the potential protective role of TPM3 and HDAC1 inhibitor MGCD0103 against hypoxia-induced damage. Our findings shed light on the intricate mechanisms underlying cardiomyocyte response to hypoxia and highlight the therapeutic potential of TPM3 modulation in mitigating hypoxia-associated cardiac injury.

CoCl_2_ has been widely used in numerous studies to create a chemical hypoxia cell model, simulating the biological state of cells under low oxygen conditions [58,59]. In this study, cells were treated with sugar-free DMEM medium supplemented with a concentration of 400 μM CoCl_2_ for 5 h. After treatment, the expression levels of HIF-1α, a classical hypoxia marker [60,61], in cells were measured to validate the modeling effect of chemical hypoxia on cardiomyocytes. Consistent with the previous literature, under our oxygen deprivation intervention, HIF-1α was significantly upregulated in cardiomyocytes. This result indicates that using hypoxia culture medium intervention can effectively induce hypoxia in cardiomyocytes, confirming the successful construction of a cardiomyocyte hypoxia model.

Moreover, significant morphological alterations and cellular damage were observed in cardiomyocytes under hypoxia conditions. Detailed morphological alterations included irregular morphology and detachment from the substrate, reflecting the detrimental impact of oxygen deprivation on cellular homeostasis. Additionally, observations of the cell cytoskeleton using immunofluorescence labeling under confocal microscopy showed that under normoxia conditions, cardiomyocyte cytoskeletal arrangement was orderly and structurally clear, while under hypoxia conditions, the fluorescence intensity of the cell cytoskeleton significantly decreased, indicating structural blurring and fragmentation of the cytoskeleton. This phenomenon suggests that hypoxia may lead to the disassembly and disruption of the cardiomyocyte cytoskeletal structure. The disruption of cardiomyocyte morphology and cytoskeletal disarray underscores the importance of oxygen availability in maintaining structural integrity and function within the myocardium.

The LDH release assay is a technique that utilizes an enzyme-catalyzed colorimetric reaction to detect the activity of LDH in samples. When cell membranes are damaged, enzymes including the relatively stable LDH are released into the culture medium. By measuring the activity of LDH in the culture medium, the toxicity of drugs and the extent of cell injury can be accurately assessed. Oxidative stress imbalance plays a significant role in myocardial cell ischemia and hypoxia injury. MDA, as a common product of lipid oxidation in the body, is often used to evaluate the level of intracellular oxidative stress. SOD, as an important antioxidant enzyme in the body, catalyzes the dismutation reaction of superoxide anions, thereby inhibiting the occurrence of oxidative stress [62]. Therefore, MDA and SOD are commonly used as key indicators to assess oxidative stress [63,64]. Our experimental results indicate that under hypoxia conditions, cell injury was significant in all groups, and oxidative stress levels increased significantly, while the antioxidant capacity of cells decreased. TPM3 overexpression attenuated hypoxia-induced morphological changes and cellular damage, suggesting a crucial role for TPM3 in preserving cardiomyocyte structural integrity under hypoxia conditions. Additionally, TPM3 overexpression mitigated hypoxia-induced oxidative stress, as evidenced by reduced levels of LDH release and MDA, along with maintained SOD activity. These findings suggest that TPM3 plays a crucial role in preserving cardiomyocyte viability and function under hypoxia stress conditions.

Similarly, through TPM3 knockdown, we observed that cell injury was further exacerbated under hypoxia conditions, oxidative stress levels increased, and the antioxidant capacity of cells decreased further, consistent with the trend observed in previous experiments. Furthermore, intervention with MGCD0103 alleviated the exacerbation of cell hypoxia injury and oxidative stress imbalance caused by TPM3 knockdown. This suggests that MGCD0103 may exert a protective effect on cardiomyocyte hypoxia injury by inhibiting the expression of HDAC1, promoting histone acetylation levels, and subsequently enhancing TPM3 transcription.

The protective mechanisms of TPM3 against hypoxia injury in cardiomyocytes were investigated through PPI network analysis and GO/KEGG functional enrichment analysis. The results highlighted involvement in biological processes such as cardiac muscle tissue development and heart contraction, as well as key pathways, including adrenergic signaling in cardiomyocytes and cardiac muscle contraction. These findings underscore the potential of TPM3 modulation as a therapeutic strategy for ischemic heart disease and other hypoxia-related cardiac conditions, emphasizing the need for further research to elucidate the specific molecular mechanisms underlying TPM3-mediated cardio-protection and validate its therapeutic potential in vivo and clinically.

In light of the findings presented, the clinical implications of TPM3 modulation in IHD treatment warrant further exploration. Given the challenge of IRI in patients with IHD, including complications like the no-reflow phenomenon [65], “stone heart” [66], and calcium desensitization post-hypothermic ischemia-reperfusion [67], strategies aimed at enhancing TPM3 expression or function could present a promising avenue to attenuate myocardial damage during acute ischemic events and subsequent reperfusion therapy. TPM3 stabilization of the cytoskeleton may maintain endothelial integrity and microvascular function, crucial for preventing microvascular obstruction and ensuring adequate perfusion following ischemic events. Additionally, TPM3 may help preserve myocardial structure and function and maintain contractility under ischemic conditions, potentially alleviating the severe damage observed in “stone heart” cases. Furthermore, by supporting proper calcium signaling and myofilament interaction, TPM3 could enhance myocardial contractility post-reperfusion, addressing calcium desensitization challenges. Note that further research is needed to validate these conjectures. Additionally, future clinical trials are warranted to evaluate the efficacy and safety of TPM3-targeted therapies in patients with IHD.

Despite the insights provided by our study, it is important to acknowledge the limitations inherent in our experimental approach. Specifically, while the use of CoCl_2_-induced hypoxia offers advantages in terms of experimental control and feasibility, it may not fully recapitulate the dynamic and multifaceted nature of hypoxic conditions encountered in vivo. Therefore, caution should be exercised in extrapolating our findings directly to clinical scenarios without further validation.

In summary, the potential mechanisms by which TPM3 protects cardiomyocytes against hypoxia injury may involve its regulation of the cytoskeleton stability. TPM3, a key regulator within the cytoskeleton, modulates actin filament dynamics and stability, influencing cellular morphology and signal transduction pathways [38]. Under pathological conditions such as myocardial ischemia and hypoxia, alterations in the cytoskeleton occur, contributing to cardiovascular diseases [42]. TPM3 has been shown to play a crucial role in regulating actin filament stability [39,40,41], which could affect cellular responses to hypoxia stress, potentially mitigating myocardial hypoxia injury [38,68].

## 4. Materials and Methods

### 4.1. Cell Culture

The H9c2 cell line and AC16 cell line were obtained from the Cell Bank of the Chinese Academy of Sciences (Shanghai, China). Cardiomyocytes were resuscitated from cryovials by rapid thawing in a 37 °C water bath, followed by transfer to centrifuge tubes and resuspension in complete medium (high glucose DMEM supplemented with 10% fetal bovine serum) (DMEM: Gibco, MA, USA) (FBS: ABW, Shanghai, China). Then, cells were seeded into culture flasks and incubated overnight. Subsequent maintenance involved regular observation and medium changes, while cell passaging was performed when cells reached 80% confluency.

### 4.2. Lentiviral Transduction for TPM3 Overexpression or Knockdown

Lentivirus was designed and generated by GenePharma (Suzhou, China). Cells were transduced with lentiviral particles. Puromycin (Solarbio, Beijing, China) selection was employed to enrich cells with stable integration of the lentiviral vector. The efficiency of TPM3 knockdown or overexpression was confirmed by Western blot. The established stable cell lines were subsequently utilized for further experimental analyses.

### 4.3. Construction of the Chemical Hypoxia Model and HDAC1 Inhibition in Cardiomyocytes

H9c2 cardiomyocytes and AC16 cardiomyocytes were treated with sugar-free DMEM medium supplemented with 400 μM concentration of CoCl_2_ (Sigma, St. Louis, MO, USA) for 5 h. After treatment, the expression levels of HIF-1α in the cells were further measured to validate the modeling effect of chemical hypoxia on cardiomyocytes. As for HDAC1 inhibition, MGCD0103 (Abmole, TX, USA) was applied to the cells at a concentration of 0.5 μM for 24 h.

### 4.4. Western Blot Analysis

Proteins were extracted from cardiomyocytes using RIPA buffer (Solarbio, Beijing, China) supplemented with protease and phosphatase inhibitors (Solarbio, Beijing, China). Cells were lysed on ice for 30 min and then centrifuged at 12,000 rpm for 15 min at 4 °C. The supernatant was collected, and protein concentration was determined using a BCA protein assay kit (Solarbio, Beijing, China). Equal amounts of protein (30 μg) were separated by SDS-PAGE and transferred onto PVDF membranes. Membranes were blocked with 5% non-fat milk in TBST (Servicebio, Wuhan, China) for 1 h at room temperature and then probed with primary antibodies overnight at 4 °C. After washing in TBST, membranes were incubated with HRP-conjugated secondary antibodies for 1 h at room temperature. Protein bands were visualized using an enhanced chemiluminescence detection system (Servicebio, Wuhan, China). GAPDH (Proteintech, Wuhan, China) was used as a loading control. HIF-1α (Proteintech, Wuhan, China) was used to evaluate the effect of hypoxia modeling.

### 4.5. Immunofluorescence Multicolor Staining

To assess the impact of chemical hypoxia and various interventions on cardiomyocyte cytoskeleton, immunofluorescence staining was conducted to detect the actin cytoskeleton and related proteins. Cells were seeded onto coverslips and cultured until reaching optimal growth conditions, followed by treatment and fixation with 4% paraformaldehyde (Servicebio, Wuhan, China). After permeabilization, cells were blocked with 3% BSA (Servicebio, Wuhan, China) and then incubated with primary antibodies overnight at 4 °C. Subsequently, cells were incubated with secondary antibodies and phalloidin (Servicebio, Wuhan, China) working solution. DAPI staining was used for nuclear visualization. Coverslips were mounted using a fluorescence-quenching mounting medium, and images were captured using a laser-scanning confocal microscope.

### 4.6. CCK-8 Assay

Cells were seeded into 96-well plates at a density of 5000 cells per well and incubated overnight. Subsequently, cells were subjected to the respective interventions according to the experimental groups. After the interventions, samples were collected at specific time points (0, 1, 2, and 3 days) for CCK-8 detection. Fresh medium supplemented with CCK-8 solution (Abmole, Houston, TX, USA) was added to replace the culture medium, followed by 2 h incubation. Absorbance measurements were conducted at 450nm using a microplate reader.

### 4.7. Measurement of LDH MDA and SOD

The LDH release, SOD activity, and MDA content were assessed as indicators of oxidative stress through colorimetric assays. The supernatant of the culture medium was collected for LDH release assessment, while the cells were utilized for SOD and MDA detection. LDH levels were determined using a commercially available LDH assay kit (Nanjing Jiancheng, Nanjing, China) following the manufacturer’s protocol. SOD activity and cellular MDA content were measured using SOD assay kits and malondialdehyde assay kits, respectively, both obtained from Nanjing Jiancheng.

### 4.8. PPI Network Analysis and GO/KEGG Enrichment Analysis

To analyze the related molecules of TPM3 and construct a PPI network, we utilized the GeneMANIA and STRING databases. The obtained molecules were further subjected to GO/KEGG enrichment analysis using R (version 4.2.1) and the clusterProfiler package (version 4.4.4). The input molecule list underwent ID conversion using the org.Hs.eg.db package (version 3.10.0), and enrichment analysis was performed to identify enriched GO terms and KEGG pathways associated with TPM3.

### 4.9. Statistical Analysis

Data were analyzed using a two-way analysis of variance (ANOVA). Statistical significance was set at *p* < 0.05. All analyses were performed using GraphPad Prism software (version 8.3.0, San Diego, CA, USA).

## 5. Conclusions

Our findings demonstrate that TPM3 plays a critical role in protecting cardiomyocytes against hypoxia-induced injury and oxidative stress. These findings provide a foundation for further research aimed at unraveling the therapeutic potential of TPM3 modulation in the context of ischemic heart disease and other hypoxia-related cardiac pathologies.

## Figures and Tables

**Figure 1 ijms-25-06797-f001:**
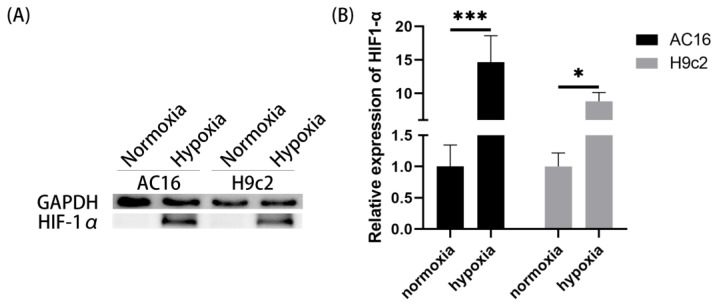
Expression of HIF-1α under normoxia or hypoxia in AC16 and H9c2 cardiomyocytes. (**A**) Representative WB images. (**B**) Quantitative analysis of the WB results (*, *p* < 0.05; ***, *p* < 0.001).

**Figure 2 ijms-25-06797-f002:**
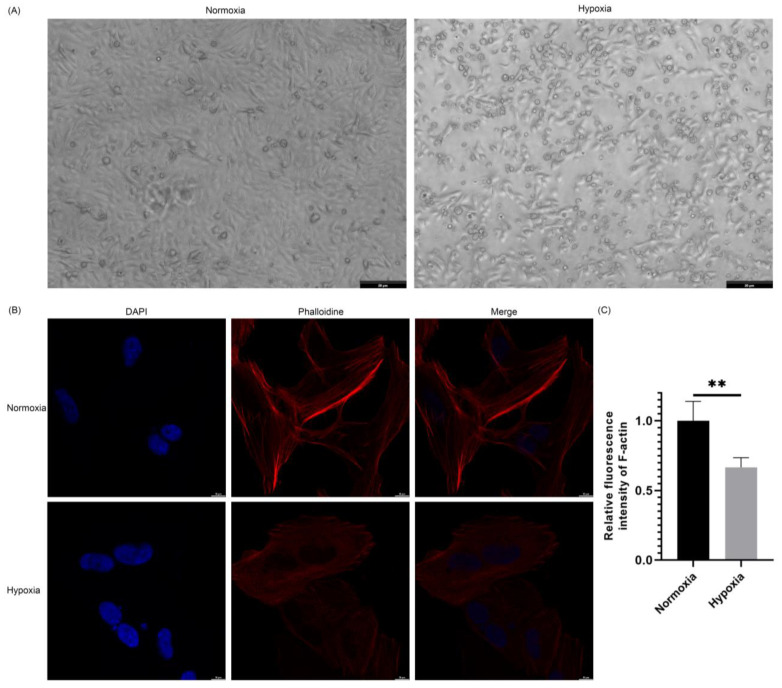
Morphological and cytoskeletal changes in AC16 cells under normoxia and hypoxia. (**A**) Phase contrast microscopy images of AC16 cells under normoxia and hypoxia conditions. (**B**) Confocal microscopy images showing cytoskeletal structure changes in AC16 cells under normoxia and hypoxia. DAPI (blue) labeling of cell nuclei and phalloidin (red) labeling of F-actin. (**C**) Quantification of the relative fluorescence intensity of F-actin (**, *p* < 0.01).

**Figure 3 ijms-25-06797-f003:**
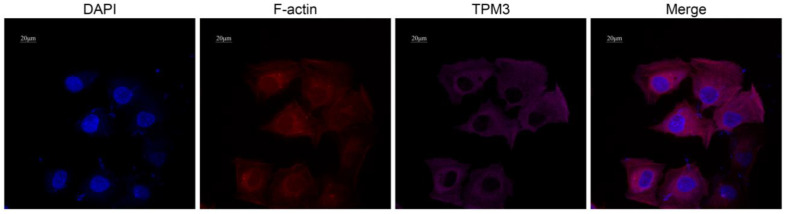
Confocal microscopy images showing the intracellular localization of TPM3 in AC16 cardiomyocytes. DAPI (blue) labeling of cell nuclei, phalloidin (red) labeling of F-actin, and TPM3 primary antibody along with Cy5 fluorescent secondary antibody (pink) labeling of TPM3 protein.

**Figure 4 ijms-25-06797-f004:**
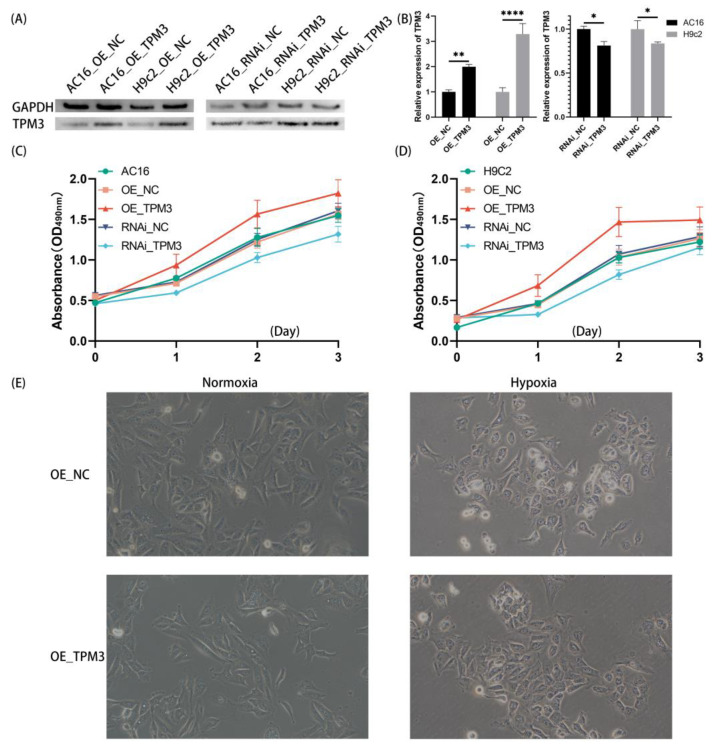
Effects of TPM3 on cardiomyocyte growth and morphology. (**A**) Western blot experiments were conducted to validate the efficacy of stable transfection in AC16 and H9c2 cells. (**B**) The quantification of the WB results. (**C**) The effect of TPM3 expression changes on the growth rate of AC16 cells was assessed using a CCK-8 assay. (**D**) The effect of TPM3 expression changes on the growth rate of H9c2 cells was assessed using a CCK-8 assay. (**E**) Phase contrast microscopy images (200× magnification) to observe the growth and morphological changes of AC16 cells (*, *p* < 0.05; **, *p* < 0.01; ****, *p* < 0.0001).

**Figure 5 ijms-25-06797-f005:**
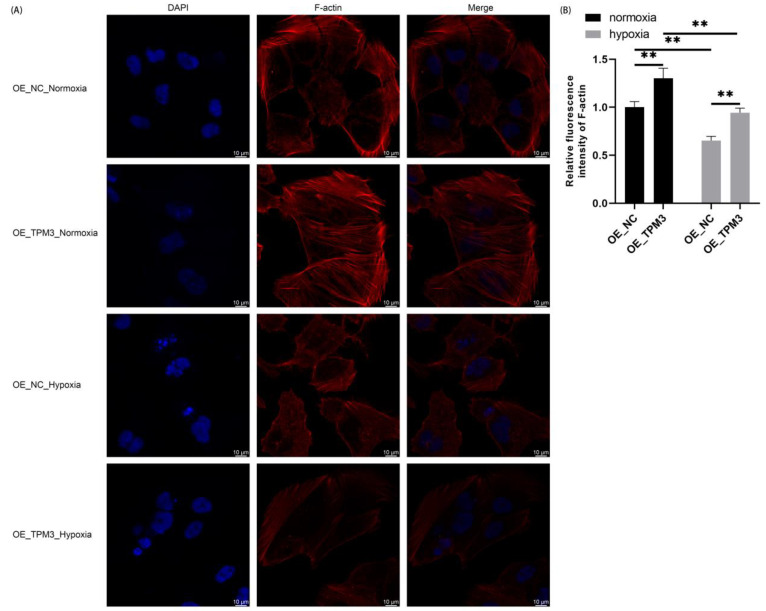
Influence of TPM3 overexpression on the cytoskeleton of AC16 under normoxia and hypoxia. (**A**) Confocal microscopy was used to observe the cytoskeleton of AC16. DAPI (blue) labeling of cell nuclei and phalloidin (red) labeling of F-actin. (**B**) Quantification of the relative fluorescence intensity of F-actin (**, *p* < 0.01).

**Figure 6 ijms-25-06797-f006:**
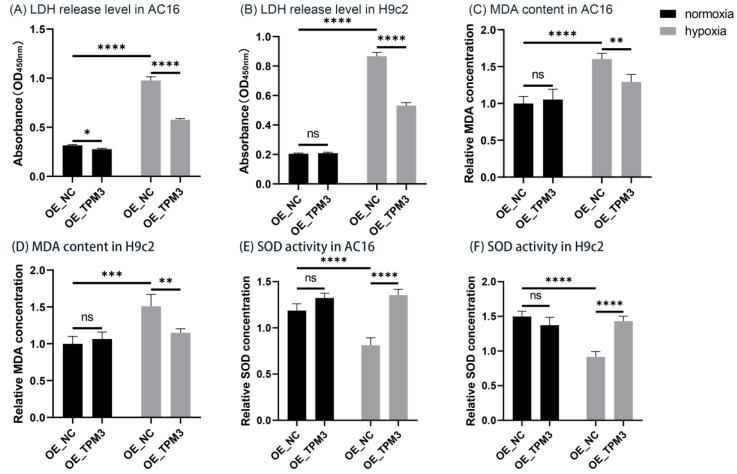
Assessment of cell injury and oxidative stress in AC16 and H9c2 cells. (**A**,**B**) LDH assay was performed to assess AC16 or H9c2 cell injury in the control group (OE_NC) and TPM3 overexpression group (OE_TPM3) under normoxia and hypoxia conditions. (**C**,**D**) MDA content detection experiment was conducted to assess the intracellular MDA content in AC16 or H9c2 cell line. (**E**,**F**) SOD activity assay was conducted to evaluate the intracellular SOD activity in AC16 or H9c2 cell line (ns, no significance; *, *p* < 0.05; **, *p* < 0.01; ***, *p* < 0.001; ****, *p* < 0.0001).

**Figure 7 ijms-25-06797-f007:**
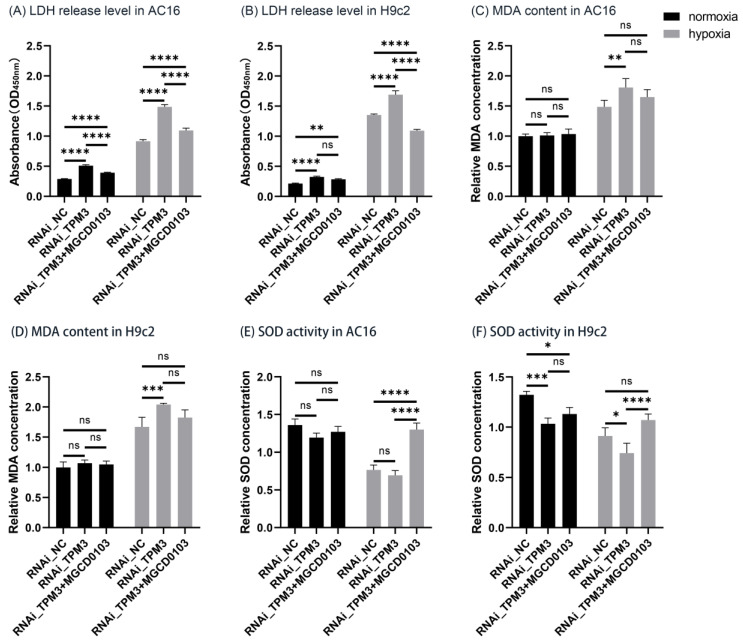
Effects of TPM3 knockdown and MGCD0103 treatment on cell injury and oxidative stress in AC16 and H9c2 cells. (**A**,**B**) LDH assay was performed to assess cell injury in AC16 or H9c2 cells in the control group (RNAi_NC), TPM3 knockdown group (RNAi_TPM3), and TPM3 knockdown group under MGCD0103 treatment (RNAi_TPM3 + MGCD0103) and under normoxia and hypoxia conditions. (**C**,**D**) MDA content detection experiment was conducted to assess the intracellular MDA content in AC16 or H9c2 cell line. (**E**,**F**) SOD activity assay was conducted to evaluate the intracellular SOD activity in AC16 or H9c2 cell line (ns, no significance; *, *p* < 0.05; **, *p* < 0.01; ***, *p* < 0.001; ****, *p* < 0.0001).

**Figure 8 ijms-25-06797-f008:**
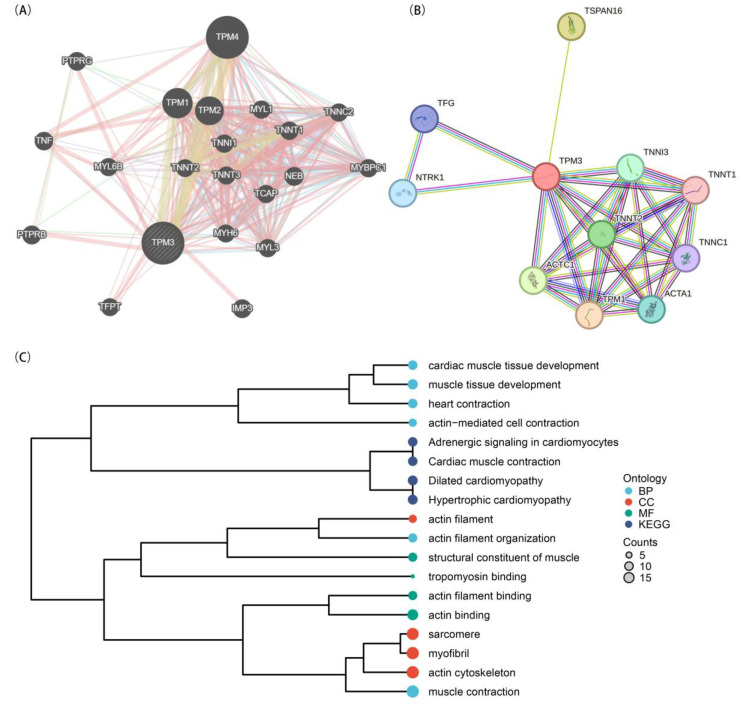
Protein–Protein Interaction (PPI) Network Analysis and GO/KEGG Enrichment Analysis of TPM3 and its related proteins. (**A**) PPI network analysis using the GeneMANIA database. (**B**) PPI network analysis using the STRING database. (**C**) GO/KEGG enrichment analysis.

## Data Availability

Public data used in this research were achieved from the GeneMANIA database: https://genemania.org/ (assessed on 1 May 2024) and the STRING database: https://cn.string-db.org/ (assessed on 1 May 2024).

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
