# Peer review of "The Role of TPM3 in Protecting Cardiomyocyte from Hypoxia-Induced Injury via Cytoskeleton Stabilization"

_ijms, 2024, doi:10.3390/ijms25126797_

Round 1

Reviewer 1 Report

Comments and Suggestions for Authors

Dear Dr. Huang,

the present paper presents original results about the role of tropomyosin 3 in ischemia-reperfusion injury. The model is a chemical-induced hypoxia cell colture which reproduce ischemia-reperfusion myocardial injury. The latter is accurately described and the conclusions are supported by well presented results. 

I would only suggest you to discuss more deeply the clinical perspectives of your results.

-The the role of tropomyosin 3 dysregulation could partially explain the no-reflow phenomenon and the "stone heart"?

-Tropomyosin 3 dysregulation might explain calcium desensitization after hypothermic ischemia-reperfusion (please see DOI:10.1186/cc13071. 

Author Response

Thank you for your kind review and constructive suggestions on our manuscript. We have carefully considered your suggestions and have expanded the discussion to address the clinical perspectives of our findings in greater depth.

Reviewer 2 Report

Comments and Suggestions for Authors

1. The title can be more specific

2. Describe what has been done but don’t discuss result in the figure legend. All figure legend will need a short title.

3. Need to improve the flow of the second paragraph of the introduction. The description of previous findings is too vague.

4. The rationale to study HDAC is weak. HDAC and TPM could just act in parallel. 

5.Figure 1 will need quantification.

6.Figure 2, intensity of actin fluorescence signal, cell volume will need to be quantified.

6. Phenotype in figure 4 lacks quantification. Cells in OE_TPM3 group under normoxia look longer than the OE_NC group, but the authors claim no difference.

7. It would be really informative if the authors can validate the key findings such as figure 5 in a hypoxia chamber, since chemical-induced hypoxia is different from oxygen deprivation.

Comments on the Quality of English Language

No severe grammar issue, but the scientific writing can be improved.

Round 2

Reviewer 2 Report

Comments and Suggestions for Authors

The authors have solved all the concerns raised during the previous round of review. No further comments.

Comments on the Quality of English Language

Please address the writing issue raised in the iThenticate report regarding wording duplication.